# Transcriptomic Changes in Cisplatin-Resistant MCF-7 Cells

**DOI:** 10.3390/ijms25073820

**Published:** 2024-03-29

**Authors:** Araceli Ruiz-Silvestre, Alfredo Garcia-Venzor, Gisela Ceballos-Cancino, José M. Sánchez-López, Karla Vazquez-Santillan, Gretel Mendoza-Almanza, Floria Lizarraga, Jorge Melendez-Zajgla, Vilma Maldonado

**Affiliations:** 1Laboratorio de Epigenetica, Instituto Nacional de Medicina Genomica (INMEGEN), Ciudad de Mexico 14610, Mexico; araceli_silvr@ciencias.unam.mx (A.R.-S.); cienciaflosan@gmail.com (J.M.S.-L.); gmendoza@inmegen.edu.mx (G.M.-A.); fjlizarraga@inmegen.gob.mx (F.L.); 2Posgrado en Ciencias Biomédicas, Universidad Nacional Autónoma de Mexico (UNAM), Ciudad de Mexico 04510, Mexico; 3Department of Life Sciences, Ben-Gurion University of the Negev, Beer Sheva 8410501, Israel; agarciavenzor@gmail.com; 4Laboratorio de Genomica Funcional del Cancer, Instituto Nacional de Medicina Genomica (INMEGEN), Ciudad de Mexico 14610, Mexico; gceballos@inmegen.gob.mx (G.C.-C.); jmelendez@inmegen.gob.mx (J.M.-Z.); 5Laboratorio de Innovación en Medicina de Precisión, Instituto Nacional de Medicina Genomica (INMEGEN), Ciudad de Mexico 14610, Mexico; kivs09@gmail.com; 6Consejo Nacional de Ciencia y Tecnologia, Ciudad de Mexico 03940, Mexico

**Keywords:** cisplatin, MCF-7 cells, breast cancer, RNAs, transcriptome, resistance, vesicles

## Abstract

Breast cancer is a leading cause of cancer-related deaths among women. Cisplatin is used for treatment, but the development of resistance in cancer cells is a significant concern. This study aimed to investigate changes in the transcriptomes of cisplatin-resistant MCF7 cells. We conducted RNA sequencing of cisplatin-resistant MCF7 cells, followed by differential expression analysis and bioinformatic investigations to identify changes in gene expression and modified signal transduction pathways. We examined the size and quantity of extracellular vesicles. A total of 724 genes exhibited differential expression, predominantly consisting of protein-coding RNAs. Notably, two long non-coding RNAs (lncRNAs), NEAT1 and MALAT, were found to be dysregulated. Bioinformatic analysis unveiled dysregulation in processes related to DNA synthesis and repair, cell cycle regulation, immune response, and cellular communication. Additionally, modifications were observed in events associated with extracellular vesicles. Conditioned media from resistant cells conferred resistance to wild-type cells in vitro. Furthermore, there was an increase in the number of vesicles in cisplatin-resistant cells. Cisplatin-resistant MCF7 cells displayed differential RNA expression, including the dysregulation of NEAT1 and MALAT long non-coding RNAs. Key processes related to DNA and extracellular vesicles were found to be altered. The increased number of extracellular vesicles in resistant cells may contribute to acquired resistance in wild-type cells.

## 1. Introduction

Cancer continues to pose a profound challenge to global public health. According to data compiled by the World Health Organization, 9,894,402 deaths were attributed to cancer, and an estimated 18,094,716 new cases were diagnosed globally in 2020, impacting individuals of all genders [1]. Notably, breast cancer stands as the foremost cause of mortality among women, attributable to malignancy [2]. The most recent report from the World Health Organization, published in 2020, documented approximately 22 million new cases and 685,000 deaths [3].

Antineoplastic resistance stands as one of the primary culprits contributing to challenging tumor eradication, recurrence, and ultimately, mortality [4], which are among the main factors associated with death from cancer.

Some of the most commonly used antineoplastic drugs are those that target the DNA molecule by causing damage to its nitrogenous bases. This damage inhibits DNA replication, subsequent transcription, and protein synthesis [3]. This category of antineoplastic agents can exert their effects at any stage of the cell cycle, particularly targeting rapidly dividing cells. They act on the cell cycle itself or on the signaling pathways that regulate cell proliferation [4]. One prominent example of such antineoplastic agents is cisplatin ([Pt(NH_3_)_2_Cl_2_]), which serves as a primary treatment option for various cancer types like leukemia, lymphomas, lung, esophageal, testicular, prostate, ovarian, head and neck, cervical, bladder, sarcomas, neuroblastoma and breast cancers, among others [2,5,6]. Cisplatin enters the cell through passive diffusion and preferentially interacts with oxygen, nitrogen, and sulfur found in amino acids. Specifically, it forms adducts with the N7 position of DNA purines, which in turn leads to cell cycle arrest and subsequent apoptosis [7].

Cisplatin is primarily utilized for advanced breast cancers [8,9,10]. However, only 16% of metastatic breast cancers exhibit a one-year survival rate advantage when treated with cisplatin regimens when compared to regimens without it [11]. This implies that the majority of cases develop resistance to cisplatin. Numerous studies have reported antitumor resistance induced by cisplatin, wherein patients initially exhibit a favorable response to treatment. However, after multiple cycles of treatment, a chemoresistant phenotype may develop; this resistance represents the primary hurdle to overcome in chemotherapeutic treatments [12]. Antineoplastic resistance is acquired through two primary mechanisms. The first is inherent in the genetic makeup of each lineage of cancer cells. The second is acquired resistance that develops following exposure to the drug and is specific to distinct subpopulations of cells within the tumor [13].

## 2. Results

### 2.1. Cisplatin-Resistant Breast Cancer Cells

To ensure the credibility of our study, we initially verified the authenticity of the MCF-7 cell line with a genetic STR profile, concluding that the sample identified as “MCF7” possesses the same 14 alleles reported for the human MCF7 cell line (ATCC^®^ HTB-22TM) (SF1). With this information, it can be concluded that the “MCF7” sample contains only cells from the afore-mentioned cell line. There was no evidence that the analyzed sample contained more than two types of human-origin cells; in other words, the sample is not contaminated with other types of human cells (Appendix A). Then, we induced cisplatin resistance in MCF-7 breast cancer cells by gradually exposing them to escalating sublethal concentrations of the drug. The initial concentration was 0.375, and the final concentration reached 1.9 μM; higher concentrations induced massive cell death. Subsequently confirmed cisplatin resistance in cpR-MCF-7 cells through viability assessments. The obtained data unveiled significant differences in IC50 values (18.53 ± 0.63 and 33.58 ± 6.37) and IC75 values (7.9 ± 0.34 and 14.97 ± 3.98) between wt-RMCF-7 and cpR-MCF-7, respectively (Figure 1A,B), solidifying the reliability of our chosen biological model. With cisplatin resistance confirmed, we explored potential variations in the transcriptome associated with this resistance.

The principal component analysis (PCA) graph illustrates the distribution of data points in a reduced-dimensional space (Figure 1C). It clearly delineates two distinct experimental groups in the transcriptome analysis, where the wild-type cell data and the data from resistant cells are distributed in different spaces, indicating that we indeed have two experimental groups. The cpR-MCF-7 group exhibits modulation in the expression of 724 genes compared to wt-MCF-7 cells, as depicted in the volcano plot. Some RNAs show a decrease in expression levels, while another group of RNAs demonstrates an increase (Figure 1D). We applied specific cut-off values: Log2 FCh ≥ ±1 and an adjusted *p*-value ≤ 0.05. We observed that 469 RNAs exhibited increased expression, while 256 RNAs showed decreased expression compared to wt-MCF-7 cells, as depicted in Figure 2A. We want to emphasize that in this section that a subset of breast cancer cells resistant to cisplatin was successfully established, and this phenotype was accompanied by the dysregulation of their transcriptome. The quantity and quality of RNA obtained from cell cultures showed a good amount and very good purity and integrity. The next-generation sequencing ran smoothly, demonstrating good depth and quality (SF2).

### 2.2. Changes in the Transcriptome of Cisplatin-Resistant MCF7 Cells

In the group of 724 mRNAs, the majority (>95%, 692) corresponds to messenger RNA (mRNA), while 20 are pseudogenes, 7 are long non-coding RNAs (lncRNAs), and 5 are small nucleolar RNAs (snoRNAs) (Figure 2B). Out of the 692 messenger RNAs, 447 showed an increase in their expression, while 245 showed a decrease (Figure 2C). Among the seven lncRNAs, five exhibited increased expression, whereas two showed decreased expression. Similar trends were observed for pseudogenes and other RNAs (Figure 2C).

Next, to assess the clustering of the differentially expressed genes (DEGs), we generated a heatmap (Figure 2D). The results demonstrate that there is a cluster of up-regulated coding RNAs among the DEGs. Additionally, these findings indicate the presence of a specific group of genes that are preferentially upregulated in the cpR-MCF-7 cells. These results suggest the existence of regulatory differences between the experimental and control groups.

Figure 3A displays the top 10 of DEGs among protein-coding RNAs and the 7 DEGs among long non-coding RNAs (lncRNAs) (Figure 3B). The messenger RNA (mRNA) that exhibited the highest positive modification in expression was MUC5AC, while ANKRD30A showed the most significant negative modulation. Some notable lncRNAs that were differentially expressed (DE) include MALAT, DANCR, and NEAT1. In addition, FAM225A, TMEM105, LINC02875 (C17orf82), and GPC1-AS1 (PP14571) were also identified as DE lncRNAs.

These findings reveal a notable contrast in transcription patterns between cisplatin-resistant and non-cisplatin-resistant breast cancer cells, with a majority of significantly differentially regulated genes (SDRs) being protein-coding RNAs. The complete list is found in Appendix A in an Excel form (SF3). In order to validate the outcomes obtained from RNA sequencing (RNAseq), we conducted a quantitative real-time polymerase chain reaction (qRT-PCR) as an additional step (Figure 3C). This enabled us to confirm the precision of the observed alterations in cpRMCF-7 cells due to cisplatin resistance.

### 2.3. Breast Cancer Cells Associated with Cisplatin Resistance Exhibit Dysregulation in Multiple Cellular Processes

Afterward, the main deregulated cellular processes in cisplatin-resistant cells were inferred using various analyses (Webgestal, Gprofile, SDSU, Reactome, Key Pathway Advisor) based on transcriptome data. The processes that were commonly identified by all the tools are listed in Table 1 and the complete lists of genes involved in each process are presented in SF4.

As expected, we found processes associated with the regulation of DNA mechanisms such as DNA synthesis (SF5), DNA repair (SF6), and chromosome organization, possibly because cisplatin, being an antineoplastic drug, primarily targets the formation of DNA adducts. Similarly, molecules associated with the immune system showed an upregulation in response to cisplatin resistance (SF7).

As anticipated, the data analysis revealed a modification in the cell cycle process, aligning with the observation that cisplatin-resistant cells exhibit a slower rate of proliferation. In Figure 4A, it is evident that several genes associated with the cell cycle displayed altered expression patterns, marked by a decrease, including MYC, CDK4L, CCNE2, CCNA1, CCNA2, TP53INP1, RBL1, E2F1, E2F2, and SKP2. Furthermore, we identified additional upregulated genes, such as CDKN1A (p21), as illustrated in Figure 4B. Bioinformatic analyses propose that SKP2 may play a pivotal role as a hub gene in regulating the cell cycle in cisplatin-resistant MCF-7 cells (SF8).

Interestingly, we found that one of the major processes disrupted in cisplatin-resistant cells was cellular communication (Table 1). As observed in Figure 4C, diverse mRNAs related to extracellular vesicle formation were dysregulated. Our results suggest that a group of DEGs contributes to the promotion of endocytosis, which has been shown to be associated with cisplatin resistance (Figure 4D), including CD68, SPNS2, HLA-DRB5, LRP1, RAB3A, RAB43, and CHMP5, among others.

Since there is evidence suggesting that the transport of biomaterials by vesicles can contribute to tumor resistance against chemotherapy, we decided to measure the sensitivity to cisplatin in MCF7 wild-type cells cultured with conditioned medium from cisplatin-resistant cells. As shown in Figure 5A, the wt-MCF-7 cells were less sensitive to cisplatin when cultured with a conditioned medium derived from resistant cells. When we analyzing the number and size of nanoparticles in the culture medium of resistant cells and comparing them to those of wild-type cells, we found that although there was no difference in the size of nanoparticles (Figure 5B), there was a higher number of nanoparticles secreted by cisplatin-resistant cells (Figure 5C) (SV1,2). Employing Western blot, it was found that extracellular vesicles are positive for CD63 and CD9.

## 3. Discussion

Antineoplastic resistance to cisplatin in breast cancer is a significant clinical concern. Cisplatin is a commonly used chemotherapeutic agent that induces DNA damage and cell death in cancer cells. However, the development of resistance to cisplatin limits its effectiveness in treating breast cancer. After generating a subclone of cisplatin-resistant breast cancer cells through exposure to escalating doses, we obtained a cellular clone with increased antineoplastic resistance. The IC50 (half-maximal inhibitory concentration) for this clone doubled compared to the parental cells, with values ranging from 18 to 34 μM for the IC50, and 7 to 14 μM for the IC75. Upon analyzing the transcriptome using next-generation sequencing, we identified 724 differentially expressed coding RNAs in the resistant cells. Several non-coding RNAs also exhibited differential expression.

NEAT1 was the most significantly differentially expressed long non-coding RNA (LncRNA), and its elevated expression has been associated with chemoresistance to cisplatin in various malignant neoplasms, including ovarian cancer [14], medulloblastoma [15], non-small cell lung cancer (NSCLC) [16], and breast cancer [17], among others. In the context of breast cancer, XueDong Wang et al. (2021) [18]. reported the detection of NEAT1 expression in extracellular vesicles, demonstrating its association with the clinical characteristics of breast cancer patients. The study also revealed that elevated levels of NEAT1 were associated with sensitivity to cisplatin in these patients. These findings are consistent with our results and support the proposition that NEAT1 is associated with chemotherapy resistance [18]. It would be interesting in the future to verify whether, in our model, the pathway through which NEAT1 modifies drug sensitivity aligns with the one proposed by the XueDong Wang group, which involves the protein KLF12 and miR-141-3p.

MALAT1 is another long non-coding RNA (lncRNA) that has been widely reported as differentially expressed in cancer. Its overexpression is associated with cisplatin chemoresistance in various neoplasms [19,20], but not yet in breast cancer. This aspect should be explored more deeply in the future, analyzing whether the five lncRNAs (NEAT1, MALAT1, pp14571, c17orf82, and TEMEM105) found at elevated levels in cisplatin-resistant cells could potentially serve as a signature for detecting chemoresistance in breast cancer, and investigating the mechanisms of action for each of these lncRNAs.

With a comprehensive analysis of the transcriptome and differentially expressed RNAs in resistant cells, as expected, we identified alterations in processes associated with DNA regulation in cisplatin-resistant cells. Furthermore, the expression of genes associated with the regulation of the cell cycle was modified, suggesting a decreased capacity for cellular proliferation. An interesting change in cellular processes was related to cellular communication, and we observed that RNAs associated with the formation of extracellular vesicles exhibited altered expression.

An intriguing and well-documented phenomenon is the involvement of extracellular vesicles in tumor cells and their role in chemoresistance, particularly to cisplatin, across various types of tumors. Our data suggest that extracellular vesicles may also be involved in chemoresistance processes in breast cancer. We demonstrated that the conditioned medium from resistant cells imparts cisplatin resistance to wild-type cells when cultured for 42 h, implying a horizontal transfer of the chemoresistant phenotype between cells. This finding has significant implications for understanding the mechanisms of resistance to antineoplastic agents and contributes to proposing new candidates that could serve as diagnostic tests or biomarkers to aid in the detection and monitoring of high-risk patients. Interestingly, upon analyzing the extracellular vesicles, we observed that, while the size of vesicles from resistant cells does not vary significantly, there is an increase in the number of secreted vesicles. Future studies should investigate changes in the transcriptome of these extracellular vesicles could provide crucial insights into cellular communication and the transfer of genetic information, contributing to increased chemoresistance. We identified several non-coding lncRNAs, particularly noting that two of them (NEAT1 and MALAT) have existing precedents associated with chemo-resistance in cancer. NEAT1, has been detected in the exosomes of breast cancer patients, showing an association with clinical parameters, particularly metastasis and chemoresistance, is of great interest. This prompts further investigation to explore whether the presence of these lncRNA molecules in the circulation could serve as biomarkers, akin to liquid biopsy, in subsequent studies. Furthermore, it would also be of great significance to delve into the mechanism of action of MALAT in breast cancer, particularly in the phenomenon of chemoresistance. Additionally, investigating whether this lncRNA could also be present in the exosomes of breast cancer patients is crucial. In the case of NEAT1, it has already been reported that its primary mechanism is attributed to its function as a sponge for miRNAs, specifically miR-141-3p, which in turn affects the levels of the KLF12 protein [17].

## 4. Materials and Methods

### 4.1. Cell Culture

The MCF-7 cell line purchased from ATCC was cultured in RPMI 1640 with L-glutamine (Thermo Fisher Scientific, Madison, WI, USA, cat. doi:10.040CMR) at 5% SFB (Thermo Fisher Scientific, Madison, WI, USA, cat. 26140-079).

### 4.2. Generation of Subclone Resistant to Cisplatin

When MCF-7 cells reached a confluence of 90%, they were grown on 100 mm × 20 mm treated Petri dishes (Corning, NY, USA, cat. 430167) and treated with sublethal doses of cisplatin (Zuridry, Zurich Pharma, Zurich, ON, Canada, QT, MEX, cat. 04-02002 version 03), gradually increasing the concentration until the dose equivalent to clinical application was reached. The cisplatin concentrations used in the cell line MCF-7 ranged from 0.09 μM to 1.9 μM (Appendix A). Cells resistant to different cisplatin treatments were selected by selective pressure. Once the cisplatin-resistant subclone was generated at the concentration of 1.9 µM, it was intermittently exposed to the same concentrations of cisplatin to preserve drug-resistant clones and treatment ceased 6 days prior to all experiments. The cisplatin-resistant subclone (1.9 µM) was appointed cpR-MCF-7. This amount represents the maximum limit that cells can tolerate without affecting their reproductive viability.

### 4.3. Cell Viability

The cellular viability of MCF-7 cells exposed to cisplatin was inferred by MTS assays which do not measure viability directly but rather assess metabolic activity. The wt-MCF-7 and cpR-MCF-7 cells were cultivated 3.6 × 10^4 on plates of 96 wells (Ultracruz, Dallas, TX, USA, cat. sc-204447). Twenty-four hours after planting, the different concentrations of cisplatin were administered, i.e., 30 μM, 60 μM and 90 μM. Then, 39 h post-cisplatin, an MTS assay to detect viability was performed with The CellTiter 96 AQueous Non-Radioactive Cell Proliferation Assay kit and protocol (Promega, Madison, WI, USA, cat. G5430).

### 4.4. RNA Extraction

The RNA was extracted from cpR-MCF-7 and wt-MCF-7 cells using the Trizol method (Thermo Fisher Scientific, Waltham, MA, USA, cat. 15596018). Subsequently, the RNA integrity was assessed using denaturing agarose gel electrophoresis [21]. The isolated RNA was utilized for sequencing and RT-qPCR validation.

### 4.5. Transcriptome Sequencing

RNA sequencing was conducted at the Sequencing Unit of the National Institute of Genomic Medicine (INMEGEN). The integrity of the RNA samples was analyzed using capillary electrophoresis. Samples with an integrity score greater than 8.5 underwent rRNA depletion using biotinylated capture probes from the Ribo-Zero rRNA Removal Kit (Human/Mouse/Rat) (Illumina, San Diego, CA, USA). Construction of the cDNA library was performed using the TruSeq RNA All Stranded Kit (Illumina, San Diego, CA, USA) starting from 140 bp RNA fragments. The constructed library was sequenced on a Next Seq 550 System sequencer (Illumina, San Diego, CA, USA), generating paired-end reads with an average of 44 million reads per sample. The quality of the reads was assessed using FastQC (version 0.11.7) [21]. Alignment to the Homo sapiens genome reference Version GRCh 38 was carried out using STAR [13], and abundance quantification and normalization were obtained using DEseq [22]. Differential expression analysis was conducted using the R package EdgeR xrw [23].

### 4.6. qRT-PCR Validation

Nine genes from the cpR-MCF-7 cellular line were selected. The selection criteria for candidate genes for validation were homogeneity in RPKMs between biological replicates. We were looking for candidates who had obtained a fold change ≥ ±2 and a *p*-value less than 0.05 (Appendix A (ST1)). Specific oligonucleotides were designed for each of the nine candidate genes for validation. The sequence of each gene was obtained from NCBI and oligos were designed and analyzed in Primer3Plus bioinformatics software version 2.0 [24]. Contaminating DNA genomic in the samples were removed by DNase I treatment using the RQ1 RNase-Free DNAse kit and protocol (Promega, Madison, WI, USA cat, M6101). cDNA was synthesized by the Maxima First Strand cDNA Synthesis Kit for RT-qPCR, (Thermo Scientific, Madison, WI, USA, cat. #K1641) following the suppliers’ instructions. qPCR was performed SYBR^®^ Select Master Mix (Thermo Fisher Scientific, Madison, WI, USA, cat. 4472897) in the 7900HT Fast Real-Time PCR System (Thermo Fisher Scientific, Madison, WI, USA). The quantitative PCR was performed in 40 cycles, each cycle of 2 min and 50 °C, 10 min and 95 °C, 15 s at 95 °C, 30 s at the corresponding amplification temperature for each gene, 15 s at 95 °C, and 1 min at 60 °C. The relative expression was determined using the 2(-Delta C (T)) method, normalized with respect to the SDHA expression level. The experiment was carried out in triplicate.

### 4.7. HeatMap

HeatMap was designed using the Heatmapper tool [25] from the RPKMs with an FDR < 0.05. The Pearson correlation was used as the distance measurement method and centroid clustering.

### 4.8. Statistical Analysis

Data are presented with the standard deviation (SD); error bars are representative of three independent experiments (biological replicates). For statistical analysis, GraphPad (Prism) was used. The mean values of the two groups were compared using the unpaired Student *t*-test. Statistical significance was considered *p* 0.05 > (*), *p* 0.01 > (**) o *p* 0.001 > (***).

### 4.9. Functional and Pathway Enrichment Analysis

In order to infer the roles of the differentially expressed genes, Webgestalt, (http://www.webgestalt.org, accessed on 3 January 2024), Gprofile (https://biit.cs.ut.ee/gprofiler/gost, accessed on 21 January 2024), SDSU (DEP), KPA (KeyPathway Advisor) tool [26], ShinyGO 0.80 [27] with the KEGG (Kyoto Encyclopedia of Genes and Genomes) [28,29], and GSEA (Gene set enrichment analysis, 23 May 2023) [30] tools were used. Enricher (https://maayanlab.cloud/Enrichr/, accessed on 23 May 2023) application enables searching different enrichment terms for molecular function, biological process, and cellular components. To visualize the enrichment results, Appyter was used to form the graphs. To further investigate the enriched pathways associated with DEGs, Webgestalt, Gprofile, KPA, and SDSU were used. GSEA was employed with the RPKMs of each sample filtered according to FDR < 0.05. These data were loaded into the GSEA 4.2.3 application downloaded from (http://software.broadinstitute.org/gsea/index.jsp, accessed on 23 May 2023). GSEA pre-ranked analysis was used to determine whether a gene set was enriched in the cisplatin-resistant or wild-type MCF7 cells. The number of permutations was 1000-fold for each analysis according to the default weighted enrichment statistical method and a signal-to-noise metric for gene ranking. Gene sets were downloaded from the MSigDB (The Molecular Signatures Database) v6.2 database (http://software.broadinstitute.org/gsea/downloads.jsp; accessed on 23 May 2023). We used specifically ENSEMBL_human_gene. Chip as the reference gene sets. The analysis was performed using the GO and Hallmarks databases. The GSEA analysis includes four key statistics: enrichment score, of which the acronym is ES; normalized enrichment score, for which the abbreviation is NES; FDR; and *p* value. Only results with a *p* < 0.5 and FDR < 0.25 were considered.

Finally, we inferred the main deregulated cellular process and pathways in DEGs according to the signaling pathways shared by the different previous analyses. We did this using ShinyGO and KEEG.

### 4.10. Obtaining the Conditioned Medium

The wt-MCF-7 and cpR-MCF-7 cells were cultured in RPMI 1640 with L-glutamine (Corning, NY, USA, cat. doi:10.040CMR), supplemented with 5% FBS (Gibco Life Technologies, USA, cat. 26140-079). Resistant cells were propagated with 1.9 µM cisplatin (Zuridry, Zurich Pharma, QT, MEX, cat. 04-02002 version 03) for 4 consecutive days. After 48 h without drug exposure, cpR-MCF-7 cells and MCF-7 cells purchased from ATCC (wt-MCF-7) were seeded at a density of 36,000 cells/cm^2^ in 100 mm × 20 mm cell culture-treated Petri dishes (Corning, NY, USA, cat. 430167). After 24 h of seeding, an RPMI medium without FBS was applied for 26 h. After this time, the medium was collected, centrifuged at 3000× *g* for 15 min, and used as the conditioned medium for the cell viability assay. The remaining medium was stored at −70 °C for subsequent nanoparticle tracking analysis.

### 4.11. MTS Viability Assay for Cells in Co-Culture with the Conditioned Medium

The cells were seeded into 96-well plates (Ultra cruz, Dallas, TX, USA, cat. sc-204447) at a density of 36,000 cells per square centimeter. After 24 h of seeding, 250 µL of either serum-free RPMI culture medium (mF), conditioned medium from cpRMCF-7 cells (mR), or conditioned medium from wt-MCF-7 cells (mW) was added to the wells. In some wells, cisplatin was included at a concentration of 40 µM by 42 h. Following this, the MTS assay was performed using the CellTiter 96 AQueous Non-Radioactive Cell Proliferation Assay kit (Promega, Madison, WI, USA, cat. G5430) and the accompanying protocol. As of the time of writing, two experimental replicates have been carried out, with 4 and 3 technical replicates for each experimental condition.

### 4.12. Nanoparticle Tracking Assay

The size distribution and quantification of nanoparticles in the conditioned medium were analyzed using nanoparticle tracking analysis (NTA) with the NanoSight NS3000 system (Malvern Instruments, Malvern, Worcs, UK. The samples were suspended in serum-free RPMI medium, and dilutions were made at a ratio of 1:2 to achieve particle counts ranging from 2 × 10^9^ to 5 × 10^8^ per mL. Three experimental replicates were performed for each biological replicate, and two biological replicates have been conducted thus far.

### 4.13. Isolation of Exosomes and Detection of CD63 and CD9 by Western Blot

Eight million cells were seeded, and 24 h later, two washes with PBS were performed. Then, 8 mL of a serum-free RPMI medium was added for 48 h. Subsequently, the medium was collected and centrifuged for 15 min at 3000× *g* to remove cellular debris. The extraction of extracellular vesicles (EVs) was then carried out according to the instructions of the exoRNeasy Serum/Plasma Maxi Kit (cat. no. 77064, Mexico City, Mexico). Afterward, the vesicles were lysed by adding an equal volume of a strong urea-containing lysis buffer. This lysate was agitated at room temperature for 15 min on a mechanical shaker. And the total protein was obtained from MCF-7 cells collected by scraping and centrifuged for 5 min at 2000× *g* rpm. The pellet was resuspended with the Laemmli loading buffer, and the protein samples were centrifuged for 40 min and boiled for 10 min. The protein concentration was assessed using the EZQ™️ Protein Quantitation Kit (Thermo Fisher Scientific, Madison, WI, USA, cat. R-33200), and 1.7–8.0 μg of protein was loaded into 4–20% sodium dodecyl sulfate polyacrylamide gels (Lonza, Walkersville, MD, USA cat #59111). Total proteins of the MCF-7 cell line and EVs were separated by SDS-PAGE (SDS-polyacrylamide gel electrophoresis) and transferred to polyvinylidene fluoride membranes. Afterward, the membranes were blocked with 5% nonfat milk in TBS-T (TBS with 0.1% of Tween20) at room temperature for 1 h, and then incubated with the primary antibodies anti-CD63 (sc-5275, 1:200) and anti-CD9 (sc-13118, 1:200) overnight at 4 °C. Next, the membranes were washed with TBS-T and incubated with the secondary antibody anti-mouse 1:2000 (Promega, Madison, WI, USA, cat W402B) for 1 h at room temperature. After washing the membranes with TBS-T, the signals were developed with Immobilon Western Chemiluminescent HRP Substrate (Merck Millipore, Burlington, MA, USA cat WBKLS0500), according to the manufacturer’s instructions. Images were acquired using the Bio-Rad gel documentation system.

## Figures and Tables

**Figure 1 ijms-25-03820-f001:**
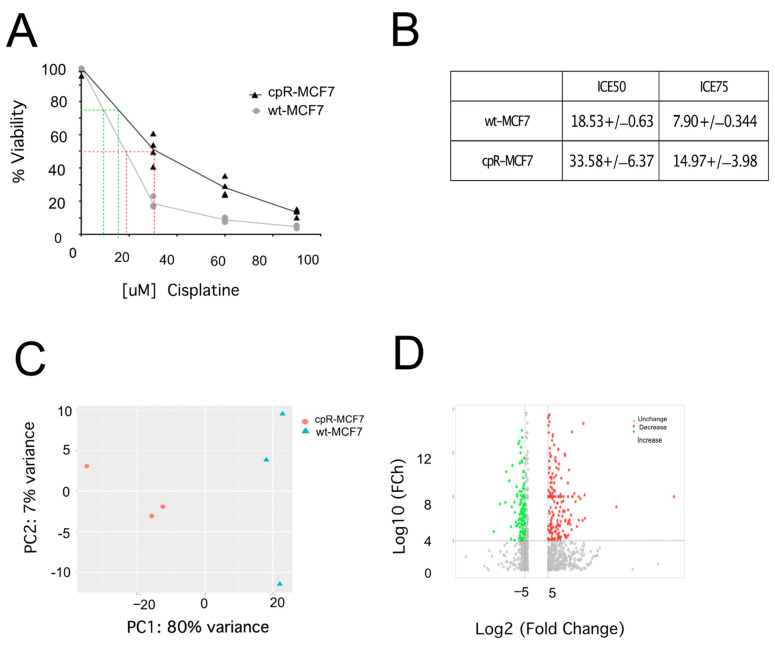
Generation of cisplatin-resistant breast cancer cells. (**A**) MTS assays were performed on cpR-MCF-7 and wt-MCF-7 cells exposed to 30, 60, or 90 µM of cisplatin for 39 h. (**B**) The maximum inhibitory concentration at which cell viability is inhibited by 50% and 75% (IC50 and IC75) was determined for cpR-MCF-7 and wt-MCF-7 cells. (**C**) Principal component analysis was conducted on RNAseq biological replicates from cpR-MCF-7 and wt-MCF-7 cell samples. (**D**) A volcano plot was generated to illustrate the distribution of unfiltered data and data filtered by *p*-value < 0.05 and FC ≥ 2. The mean of the three experimental replicates is plotted. The data were analyzed using nonparametric Student’s *t*-test. Points on the curve represent the mean values.

**Figure 2 ijms-25-03820-f002:**
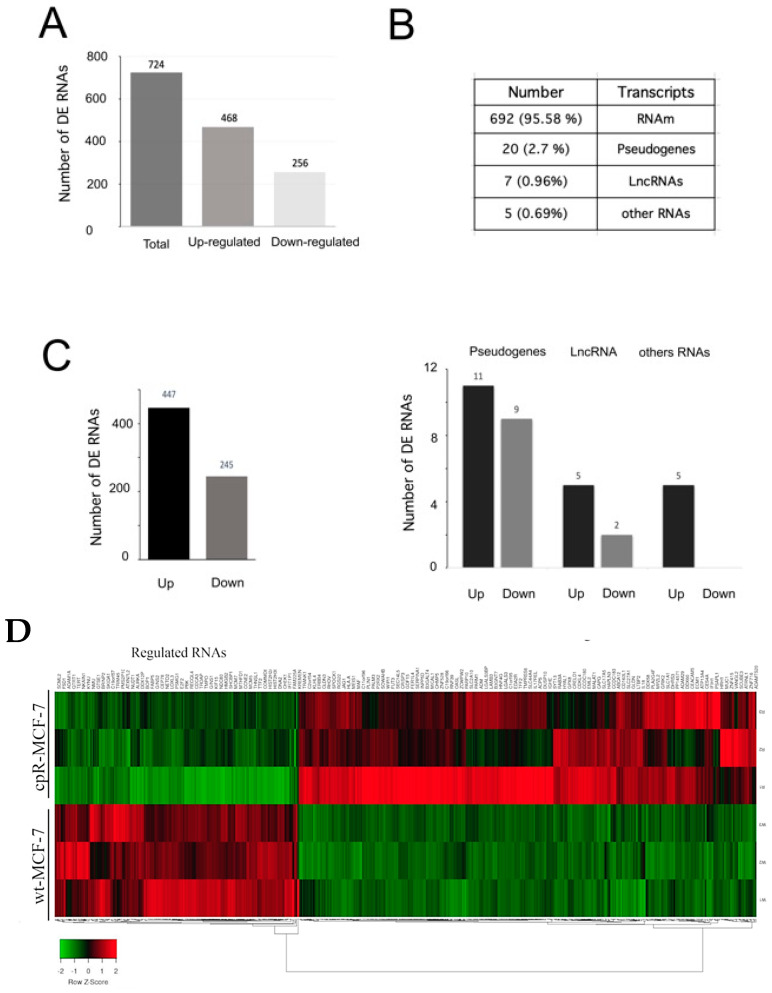
Differential gene expression analysis in cpRMCF-7 cells. (**A**) Representation of the total number of transcripts with upregulated or downregulated expression. (**B**) Count and percentage of transcripts up- or downregulated according to their classification. (**C**) Representation of the total number of mRNAs with upregulated or downregulated expression (**left** panel), Representation of the total number of pseudogenes, lncRNAs, and ncRNAs with upregulated or downregulated expression (**right** panel) (**D**) Heatmap of the 725 genes with differential expression.

**Figure 3 ijms-25-03820-f003:**
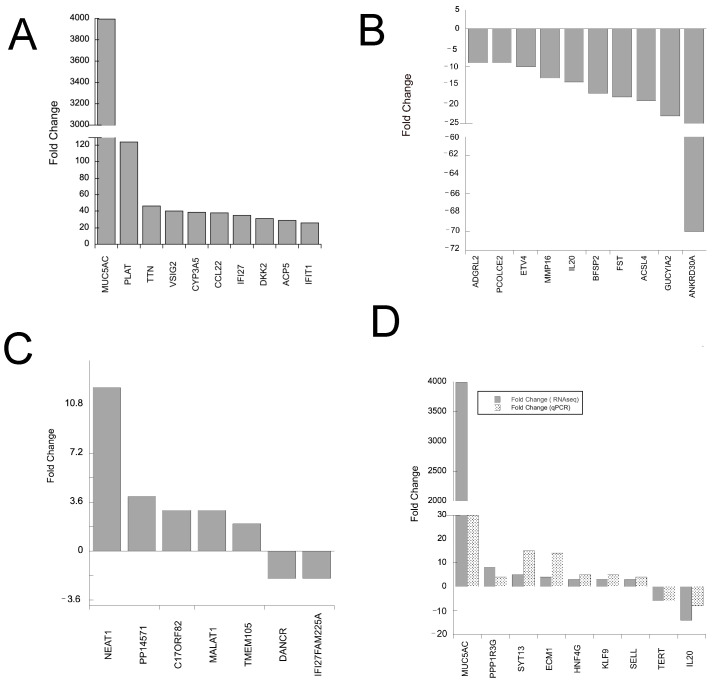
Top differentially expressed genes and validation. (**A**) Top ten upregulated mRNAs in cp-RMCF-7 cells, and (**B**) downregulated mRNAs in cp-RMCF-7 cells. (**C**) Upregulated or downregulated lncRNA expression in cpR-MCF-7 cells. (**D**) List of genes with differential expression in RNAseq validated by qRT-PCR.

**Figure 4 ijms-25-03820-f004:**
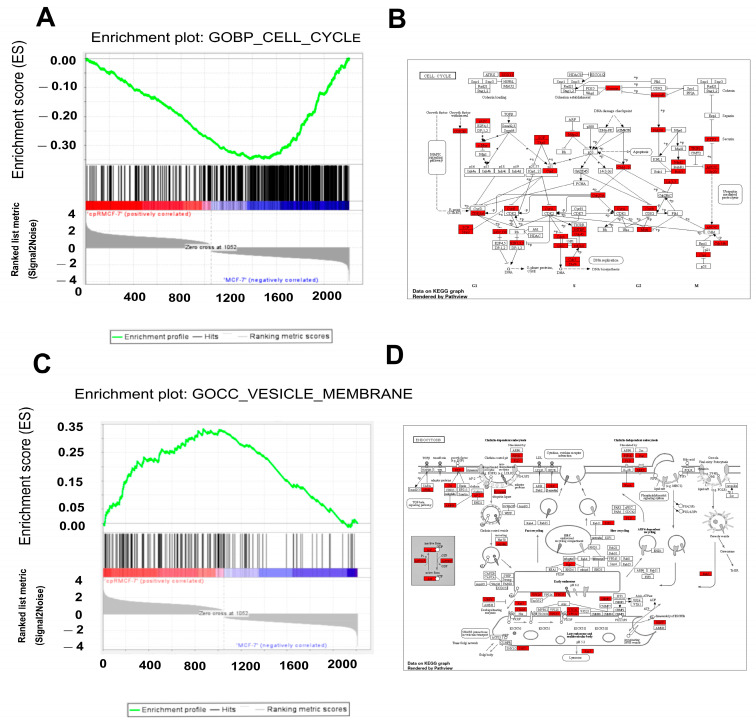
Dysregulation of the cell cycle and cell communication in cisplatin-resistant breast cancer cells. (**A**). Gene Ontology was utilized to perform gene enrichment analysis of the dysregulated genes associated with the cell cycle in cpR-MCF-7 cells. (**B**) A diagram illustrating dysregulated genes associated with the cell cycle in cpR-MCF-7 cells was generated using ShinyGO and the Kyoto Encyclopedia of Genes and Genomes. (**C**) Gene enrichment analysis of the dysregulated genes involved in cell cycle-related vesicular membrane processes was conducted using Gene Ontology in cpR-MCF-7 cells. (**D**) A diagram depicting the dysregulated genes associated with endocytosis in cpR-MCF-7 cells was generated using ShinyGO and the Kyoto Encyclopedia of Genes and Genomes.

**Figure 5 ijms-25-03820-f005:**
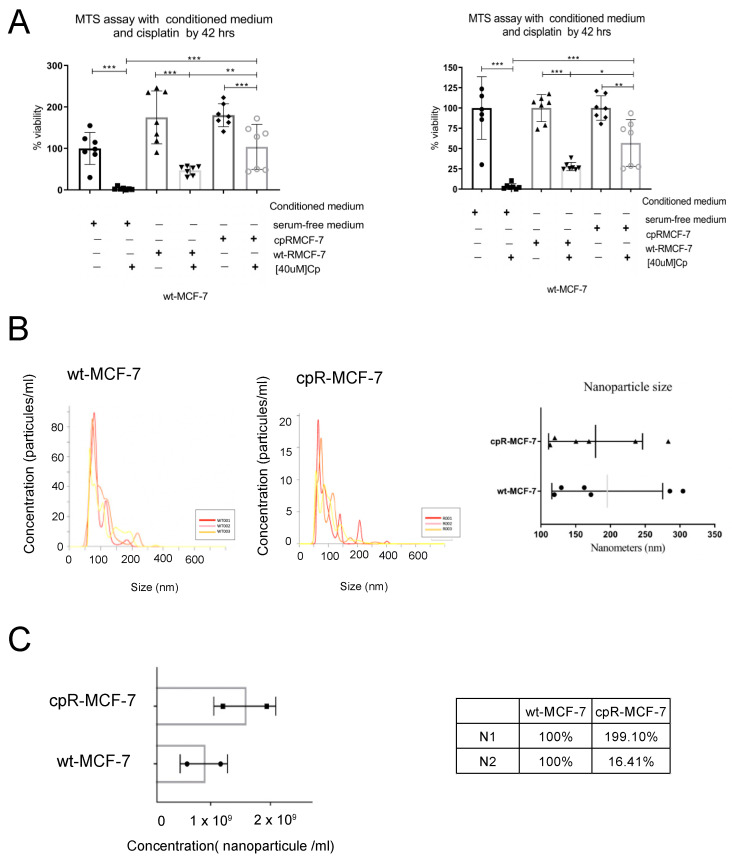
Differences exist between nanoparticles due to the cisplatin resistance effect. (**A**) The cell viability of wt-MCF-7 cells cultured with a conditioned medium from cpR-MCF-7 cells, with or without cisplatin, was evaluated for a duration of 42 h. Each plotted point represents a biological replicate. Statistical analysis was conducted using a 2-way ANOVA. Left panel: normalized groups; right panel: non-normalized groups. (**B**) Analysis of the nanoparticle tracking assay was utilized to determine the concentration of nanoparticles in the conditioned medium from wt-MCF-7 and cpR-MCF-7 cells. Each plotted point represents an experimental replicate. (**C**) Analysis of the nanoparticle tracking assay was employed to assess the size of nanoparticles in the conditioned medium from cpR-MCF7 and wt-MCF-7 cells. Each plotted point represents a biological replicate. The mean of the biological replicates is plotted with *** for *p*-value < 0.001, ** for *p*-value < 0.002, and * for *p*-value < 0.033.

**Table 1 ijms-25-03820-t001:** Inferrend cellular processes and components from cisplatin-resistant cell RNA seq data.

Processes	Adjusted *p* Value	Shared Genes	Percentaje of Shared Genes	Genes
Cell communication	1.2 × 10^8^	34	48.5	MUC5AC PLAT TTN CCL22 IFI27 DKK2 IGHE KLK5 DTX1 IFI6 BHLHA15 CD68 OAS1 PSAPL1 CXCL8 OAS2 HSH2D PPFIA2 TMPRSS6 PTPRN2 RGS22 HRH1 GLRA3 GDF15 TRIM55 NOSTRIN DDX60 FAM83E ACKR2 BTN3A1 CDKN1A ERBB4 ALCAM KLF4
HDR through Homologous Recombination (HRR) or Single Strand Annealing (SSA)	0.0003	15	75	BRCA1 CCNA1 CCNA2 CHEK1 DNA2 EXO1 PCNA POLD1 POLE POLE2 RBBP8 RFC2 RFC3 TIMELESS XRCC2
Chromosome Maintenance	0.0004	12	54.5	CENPM CENPN CENPP CENPU DNA2 PCNA POLA2 POLD1 PRIM1 RFC2 RFC3 TER
Mitotic Prometaphase	0.0017	18	90	AURKB BIRC5 CDC20 CDCA5 CDCA8 CENPM CENPN CENPP CENPU CEP78 KIF2C NCAPG NDC80 NEK2 SKA1 SPC24 SPC25 ZWINT
DNA Double-Strand Break Repair	0.0035	15	75	BRCA1 CCNA1 CCNA2 CHEK1 DNA2 EXO1 PCNA POLD1 POLE POLE2 RBBP8 RFC2 RFC3 TIMELESS XRCC2
G2/M Checkpoints	0.0176	17	73.9	BRCA1 CDC25A CDC45 CDC6 CDC7 CHEK1 DNA2 EXO1 GTSE1 MCM4 MCM5 MCM7 ORC6 PKMYT1 RBBP8 RFC2 RFC3
Base excision repair	0.0301	6	100	LIG1 NEIL3 PCNA POLD1 POLE POLE2
Mismatch repair	0.0342	6	100	EXO1 LIG1 PCNA POLD1 RFC2 RFC3
Nucleotide excision repair	0.0343	7	100	LIG1 PCNA POLD1 POLE POLE2 RFC2 RFC3
Extracellular matrix organization	0.0417	27	96.4	COL17A1 TNXB PCOLCE2 TMPRSS6 ITGA2B LOXL3 NTN4 LAMC2 KLK7 SCUBE3 EFEMP2 CAPN9 EFEMP1 SH3PXD2A CAPN5 CAPN2 TIMP1 LAMB3 HSPG2 CAPN12 MMP16 P4HA1 P4HA2
Cellular senescence	0.0563	19	100	CDKN1A CXCL8 HLA-B HLA-C PIK3R3 HLA-A HLA-F CDC25A CCNA2 CCNA1 MAPK11 RBL1 CCNE2 MYC CHEK1 CAPN2 E2F1 E2F2 MYBL2
DNA Replication	0.0855	23	92	GINS1 GINS2 RFC3 PCNA MCM7 UBE2C RFC2 PRIM1 CDC7 CDC6 CCNA2 CCNA1 POLA2 ORC6 CDC45 CCNE2 POLD1 POLE2 MCM4 MCM5 DNA2 SKP2 POLE
DNA Repair	0.1189	17	73.9	BRCA1 CCNA1 CCNA2 CHEK1 DNA2 DTL EXO1 PCNA POLD1 POLE POLE2 RBBP8 RFC2 RFC3 TIMELESS UBE2L6 XRCC2
Cell Cycle	0.1802	56	83.5	TOP2A CDKN1A MCM7 CDC14A CDC20 TUBA1B MYC NEK2 MYBL2 SKP2 GTSE1 POLE RFC3 RFC2 CDC25A CCNA2 CCNA1 RBL1 CCNE2 BIRC5 MCM4 MCM5 KIF2C DNA2 PCNA CDCA5 PRIM1 CDCA8 NCAPG HMMR PKMYT1 SKA1 AURKB AURKA ORC6 CDC45 POLD1 E2F 1 E2F2 CEP78 GINS1 CENPU GINS2 UBE2C CDC7 CDC6 NDC80 ZWINT DHFR POLA2 POLE2 CENPM CENPN CENPP SPC24 SPC25
Extracellular vesicle	0.1166	5	4.9	ARRDC4 EFEMP2 RAB3A COL6A2 PROM2

## Data Availability

Data will be made available on request.

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
