# Peer review of "Transcriptomic Changes in Cisplatin-Resistant MCF-7 Cells"

_ijms, 2024, doi:10.3390/ijms25073820_

Round 1

Reviewer 1 Report

Comments and Suggestions for Authors

This manuscript explores cisplatin resistance in breast cancer, which is a major cause of cancer-related deaths in women. The authors performed RNA sequencing, 724 genes including two long non-coding RNAs (NEAT1 and MALAT) were found to be dysregulated in cisplatin-resistant MCF7 cells analysis showed changes in DNA synthesis, cell cycle regulation, immune response, and extracellular vesicle-related processes. My comments and feedback;

·         The introduction repeats cancer-related statistics from the World Health Organization (WHO) for 2020. The authors can provide the most recent and pertinent data to maintain reader engagement.

·         A few sentences are very lengthy and complex. It could be more concise and simplify the sentence structure in the introduction.

·         Result section 2.1 lacks substantial interpretation of the results. It's important to discuss the biological implications and significance of the findings. For example, what do the changes in gene expression imply for cisplatin resistance, and how might these findings relate to breast cancer treatment? The significance of all the results outcomes is not explained.

·         There is some repetition in the text of result text 2.3, for instance, the phrase "As expected" appears more than once.

·         The experiment involving conditioned medium from cisplatin-resistant cells affecting the sensitivity of wild-type cells is interesting, but the provided information does not discuss the implications or the mechanism behind this finding.

·         The figure quality of each result section is very poor.

·         The discussion is somewhat disorganized. It would be helpful to structure the discussion into distinct subsections for each major point, such as the role of specific non-coding RNAs, alterations in cellular processes, and the implications of extracellular vesicles.

·         What do the differential expressions of NEAT1 and MALAT1 mean for cisplatin resistance in breast cancer, and what could be the potential mechanisms involved?

·         Scientific questions are interesting but there is no clear discussion and explanation of experiments, results, and discussion. Somehow readers will lose interest and flow for reading the manuscript. Computational analysis is not well done. This work really needs improvement. Further validations are required for transcriptomic changes analysis.

Comments on the Quality of English Language

Moderate editing required

Author Response

Response to Reviewer 1 Comments

Thank you sincerely for dedicating time to review this manuscript. Enclosed, you will find the comprehensive responses and the associated revisions/corrections, highlighted or tracked changes, in the resubmitted files.

 Point-by-point response to Comments and Suggestions for Authors

Comments 1: The introduction repeats cancer-related statistics from the World Health Organization (WHO) for 2020. The authors can provide the most recent and pertinent data to maintain reader engagement.

Response 1: In relation to your observation about the lack of up-to-date information on cancer statistics up to the current date, we regret to inform you that as of today, we have not found more recent data. We continue to monitor reliable sources and will remain vigilant for any relevant updates that may emerge before the final publication of the article. The World Health Organization (WHO) provides statistical data up to the year 2020 on cancer due to the time required to collect, verify, and analyze the information. The data collection for diseases, including cancer, is a time-consuming process due to the need to gather accurate and detailed information from multiple sources worldwide. Reports and statistics typically take some time to become available due to these collection and analysis processes. The wording has been slightly modified in an attempt to improve it. The introductory paragraph has been highlighted in red to facilitate the identification of the modification.

Comments 2. A few sentences are very lengthy and complex. It could be more concise and simplify the sentence structure in the introduction.

Response 2. The wording of the introduction was changed with more concise and direct sentences to improve the writing. The changes were marked in red in the introduction.

Comments 3. Result section 2.1 lacks substantial interpretation of the results. It's important to discuss the biological implications and significance of the findings. For example, what do the changes in gene expression imply for cisplatin resistance, and how might these findings relate to breast cancer treatment? The significance of all the results outcomes is not explained.

Response 3. We added some additional information in the Results section 2.1 and specified that in this phase, the crucial point is that we obtained a subclone of MCF-7 that is more resistant to cisplatin. This was confirmed through viability curves, showing a statistically significant increase in IC50 and IC75. Another significant change is that this phenotype was accompanied by a transcriptional shift, which, upon analysis, confirms that we are working with two distinct experimental groups. The implications of these changes are discussed in subsequent sections where we comprehensively analyze these alterations.

Comments 4 There is some repetition in the text of result text 2.3, for instance, the phrase "As expected" appears more than once.

Response 4. We reviewed the wording of section 2.3 and corrected any repeated words and/or phrases.

Comments 5. The experiment involving conditioned medium from cisplatin-resistant cells affecting the sensitivity of wild-type cells is interesting, but the provided information does not discuss the implications or the mechanism behind this finding.

Response 5: Additional information addressing the implications and possible mechanism has been added to the discussion section.

Comments 6 The figure quality of each result section is very poor.

Response 6: The quality of all the figures was improved to make them clear for the reader.

Comments 7.  The discussion is somewhat disorganized. It would be helpful to structure the discussion into distinct subsections for each major point, such as the role of specific non-coding RNAs, alterations in cellular processes, and the implications of extracellular vesicles.

Response 7: The discussion was restructured, and various points were developed, including the implications of lncRNAs and the direction these findings could lead to in the detection and treatment of breast cancer.

Comments 8. What do the differential expressions of NEAT1 and MALAT1 mean for cisplatin resistance in breast cancer, and what could be the potential mechanisms involved?

Response 8. In the discussion, this aspect has already been addressed—the importance of investigating whether lncRNAs could be used as biomarkers, such as liquid biopsy, given that at least NEAT1 has already been identified in the exosomes of cancer patients. In the case of MALAT, it would also be crucial to further explore the exact mechanism that allows its association with chemoresistance in breast cancer.

Comments 9

Scientific questions are interesting but there is no clear discussion and explanation of experiments, results, and discussion. Somehow readers will lose interest and flow for reading the manuscript. Computational analysis is not well done. This work really needs improvement. Further validations are required for transcriptomic changes analysis.

Response 9. We express our gratitude to the reviewer for their discerning suggestions. We concur with the observations and have diligently implemented the requisite modifications to augment the manuscript. The critiques provided by the reviewer have afforded us the opportunity to delve more profoundly into the implications of the results we had initially obtained.

Reviewer 2 Report

Comments and Suggestions for Authors

This manuscript is about a transcriptome analysis on breast cancer cell line samples. The RNA-seq analysis follows some well-regarded methods for differentially expressed genes analyses. Therefore, I do not have much concern in the overall analysis procedure. However, the material preparation is somewhat unclear. Also, this manuscript is mostly focused on the differentially expressed genes. The authors have conducted further analysis on the extracellular vesicles, but it is only touched with a very brief description about Figure 5. I think that this part is where the significance of this study lies. So, this part should be expanded with detailed descriptions and in-depth discussion.

Regarding the study material preparation, in order to generate cpR-MCF-7 cells, the authors exposed the wild-type MCF-7 to the drug at various levels of concentration. However, it is unclear at which concentration level the crP-MCF-7 were identified and extracted. If cpR-MCF-7 cell lines were extracted from various levels of drug concentration, multiple cell lines should be extracted from each concentration level to ensure enough number of biological replicates for the differential expression analysis.

The RNA-seq data analysis is at the centre of this study, but this manuscript has so little about that. Most transcriptome study papers typically include basic statistics related to the sample, sequencing and alignment quality which include (but not limited to) the total number of reads before and after QC, number of mapped reads, estimated fragment lengths, etc.

Also, the list of differentially expressed genes should be made available.

Figures and Tables need to be polished up.

Additional specific comments are as below:

Fig 1D is hard to interpret with both X and Y axes showing logFC with different bases.

In the second paragraph of page 3, it is stated that the p-value cutoff of 0.05 or 4 in negative log-scale. But, what is the base of the log-transformation?

The first sentence of the second last paragraph of page 3 says that the majority of 724 mRNA are mRNAs, which does not make sense.

In the same paragraph, it says that the differentially expressed genes include snoRNAs. Although some snoRNAs are relatively long with ~300nt in length, without the details of the RNA-seq experiment, it is hard to see whether snoRNAs are supposed to be captured in the data.

Figure 3 needs more information about the columns, especially the scale of the fold-change.

In Table 1, the p-value from the gene enrichment analysis should be accompanied by a few more information, such as the number of expressed genes and the number of differentially expressed genes for each term.

In Figure 5A, the 6th bar from the left (or the rightmost bar) clearly has two clusters of samples, with one cluster having similar viability as the samples in the 4th bar. However, the statistical test between the 4th bar and the 6th bar did not seem to have accounted for this clustering.

Figure 5A also needs the bars for cpR-MCF-7 cell lines for comparison.

Comments on the Quality of English Language

Minor spelling and grammar check.

Author Response

For research article

Response to Reviewer 2 Comments

Thank you sincerely for dedicating time to review this manuscript. Enclosed, you will find the comprehensive responses and the associated revisions/corrections, highlighted or tracked changes, in the resubmitted files.

 Point-by-point response to Comments and Suggestions for Authors

Comments 1: This manuscript is about a transcriptome analysis on breast cancer cell line samples. The RNA-seq analysis follows some well-regarded methods for differentially expressed genes analyses. Therefore, I do not have much concern in the overall analysis procedure. However, the material preparation is somewhat unclear. Also, this manuscript is mostly focused on the differentially expressed genes. The authors have conducted further analysis on the extracellular vesicles, but it is only touched with a very brief description about Figure 5. I think that this part is where the significance of this study lies. So, this part should be expanded with detailed descriptions and in-depth discussion.

Response 1. Additional information regarding the implications of the increased extracellular vesicle formation process has been added to the discussion section. Indeed, the most conclusive finding was the successful generation of a cisplatin-resistant subclone of the MCF-7 cell line through escalating dose exposure, maintaining surviving cells for several generations. The total time to achieve these subclones was approximately 8 months. Once the resistant phenotype was established and the transcriptome analyzed, a significant shift was observed involving both messenger RNAs and non-coding RNAs.

Upon analyzing the data and inferring the involved pathways and processes, it caught our attention that vesicle formation was active. This is noteworthy considering the precedent that the resistant phenotype can be horizontally transferred to other cells. This aspect was not further developed as we are currently in the process of obtaining more resistant cells. Once a clone is obtained, its proliferation decreases, and the material for experiments is limited. Additionally, we are now transitioning the analysis of extracellular vesicles to patients, hypothesizing that the analysis of NEAT1, MALAT, and MUC5AC in extracellular vesicles from blood or saliva could serve as promising liquid biopsy biomarkers for breast cancer. Although this project is still ongoing, we have incorporated some of these considerations into the discussion.

Comments 2. 

The RNA-seq data analysis is at the centre of this study, but this manuscript has so little about that. Most transcriptome study papers typically include basic statistics related to the sample, sequencing and alignment quality which include (but not limited to) the total number of reads before and after QC, number of mapped reads, estimated fragment lengths, etc.

Response 2. The graphs depicting the features and sequencing quality have been added to the supplementary materials. Overall, we encountered no issues with sequencing, and there was not much variability between samples. This is because in vitro experiments allow for well-controlled conditions, and the quality and quantity of RNA obtained for sequencing are excellent. This contrasts with the challenges of working with patient samples, which often suffer degradation during transportation from the operating room to the laboratory. Consequently, a percentage of RNA may undergo degradation.

Furthermore, by exclusively working with a single type of cells, one eliminates both intra- and intertumoral heterogeneity. Hence, there is a compelling reason to transition to patient samples in the next project.

Comments 3. Also, the list of differentially expressed genes should be made available.

Response 3 We have attached the list of differentially expressed RNAs in Excel format to the supplementary material.

Comments 4 Figures and Tables need to be polished up.

Response 4. The clarity and effectiveness of figures and tables have been enhanced.

Comments 5. Fig 1D is hard to interpret with both X and Y axes showing logFC with different bases.

Response 5. The units on the Y-axis of Figure 5D have been corrected.

Comments 6 In the second paragraph of page 3, it is stated that the p-value cutoff of 0.05 or 4 in negative log-scale. But what is the base of the log-transformation?

Response 6: The error in the legends of the component graphics and the manuscript has already been corrected.

Comments 7.  The first sentence of the second last paragraph of page 3 says that the majority of 724 mRNA are mRNAs, which does not make sense.

Response 7. The wording of the paragraph on page 3 has been corrected.

Comments 8.  In the same paragraph, it says that the differentially expressed genes include snoRNAs. Although some snoRNAs are relatively long with ~300nt in length, without the details of the RNA-seq experiment, it is hard to see whether snoRNAs are supposed to be captured in the data.

Response 8. We have already appended the general and filtered databases, and with a fold change above 2, we identified 5 snoRNA (11, 36,66,40 and 78). Additional information in Excel format is attached in the supplementary material.

Comments 9 Figure 3 needs more information about the columns, especially the scale of the fold-change.

Response 9. The graphs in Figure Three have now been converted to bar
chart format to make the scales more precise. In effect, the tables in
Excel with incorporated bars are only an approximation of the increases
or decreases of each RNA. However, the advantage is that it allows for
the full names of the RNAs, not just the symbols. So, as the Figure
Three was, we moved it to supplementary material so that if readers want
to identify the names of the RNAs, they can do so.   

Comments 10 In Table 1, the p-value from the gene enrichment analysis should be accompanied by a few more information, such as the number of expressed genes and the number of differentially expressed genes for each term.

Response 10. Information has been added to the table, and in the supplementary material, lists of genes involved in each modified cellular process have been included.

Comments 11 In Figure 5A, the 6th bar from the left (or the rightmost bar) clearly has two clusters of samples, with one cluster having similar viability as the samples in the 4th bar. However, the statistical test between the 4th bar and the 6th bar did not seem to have accounted for this clustering.

Response 11.

Comments 12. Figure 5A also needs the bars for cpR-MCF-7 cell lines for comparison.

Response 12. The graph of the normalized groups has been changed to one that includes all samples. We initially presented it this way for ease of interpretation by readers, but we agree with the reviewer that the other graph provides complete information.

Note: We appreciate the reviewer for their valuable critiques and suggestions. We have incorporated all the changes and hope that the revised version is now at the level for publication. We look forward to the reviewer's decision and any additional comments they may have.

Reviewer 3 Report

Comments and Suggestions for Authors

In this work, Ruiz-Silvestre et al. generated a cisplatin resistant breast cancer cell line and further characterized it in vitro including gene expression analysis. The results suggest that resistance could be mediated by factors/vesicles secreted by the cells.

Major points
-For the first figure and associated test, please explain if the transcriptome was sequenced from cells growing without selection (cisplatin) and why. For example, if one wanted to reveal responses to the drug challenge, one would sequence how both cells responded in that situation and not under no selective pressure conditions.

-For the second figure, please explain in E what are the replicates, biological replicates from independently generated clones or selections, technical replicas in the same experiment, etc.

-In figure 3, explain if the genes that were validated were specifically chosen and the rest could not be validated (and note those as well).

-MTS and MTT assays do not measure viability directly but metabolic activity. Unless viability was measured independently using other means, this should be corrected in the text.

-In figure 5 C, it is unclear what the identity of the bars are, but assuming that the upper bar corresponds to the cpR cells and the number of vesicles (and not the quality, size, content) is relevant for the observed effect, the authors should try to dilute the particles to equivalent levels as in normal cells and see if the effect disappears. Ideally, the authors should titrate the conditioned medium (in the presence of Cisplatin) of both wt and cpR cells to see if the effects can be simply explained (or correlate) with vesicle numbers. Ideally, the authors should do some minimal molecular characterization (Western blot, Mass spec) of those vesicles to support that conclusion (that the quantity and not the quality is important).

-Since in figure 5A, the authors are using conditioned media, to be able to attribute the effects to vesicles, and not to other soluble factors present in the media, the protective effects from media where vesicles are removed by physical means, should also be shown. Ideally, also, purified vesicles from the media should be added to non-conditioned media and being able to reproduce the effect seen in 5A.

Minor points

Please see attached text for edits.

Comments on the Quality of English Language

Please see some comments/edits in attached text

Author Response

For research article

Response to Reviewer 3 Comments

Thank you sincerely for dedicating time to review this manuscript. Enclosed, you will find the comprehensive responses and the associated revisions/corrections, highlighted or tracked changes, in the resubmitted files.

Point-by-point response to Comments and Suggestions for Authors

Comments 1: For the first figure and associated test, please explain if the transcriptome was sequenced from cells growing without selection (cisplatin) and why. For example, if one wanted to reveal responses to the drug challenge, one would sequence how both cells responded in that situation and not under no selective pressure conditions.

Response1: The analysis of the resistant phenotype was performed without cisplatin incubation, as the process of obtaining the phenotype is a lengthy one, spanning approximately 8 months. During this period, the cisplatin dose is gradually increased to select subclones that survive drug exposure without reproductive death, allowing the expansion of resistant cells. As observed in the transcriptome analysis, this clone exhibits various modified signal transduction pathways, some of which are expected, such as the machinery associated with DNA repair or proliferation processes. Unexpected and interesting pathways were also identified, such as the production of extracellular vesicles, which is addressed in this manuscript, and the NF-kappa B pathway, which is being explored collaterally. Both MCF-7 subclones respond differently to cisplatin, as evidenced by differences in sensitivity and modifications in IC50 and IC75, as well as in their proliferation capacity, with resistant cells exhibiting slower proliferation. Analyzing the transcriptome in both cell lines during drug exposure would likely reveal different responses at various points in the treatment, both at short times (minutes or hours) and longer durations (3-12 hours). It is a very interesting proposition.

Comments 2: For the second figure, please explain in E what are the replicates, biological replicates from independently generated clones or selections, technical replicas in the same experiment, etc.

Respuesta 2: During the process of obtaining the resistant subclone, each time the dose is increased, it is done in multiple boxes. Since the process is lengthy, samples are frozen at each step, while others continue to expand for treatment with the next dose. If cells die in the last dose, tubes from the immediate previous clone are thawed, and the dose is increased to a lesser extent. In this way, we end up with different boxes that have survived the last dose, which in our case was 1.9 uM.

For the execution of the cell viability experiment, once a subclone of MCF-7 cells resistant to death from constant exposure to cisplatin at a concentration of 1.9 uM was obtained, 3 experimental replicates of cpR-MCF-7 cells and wt-MCF-7 cells were paired and seeded with equal population density. This was done to determine IC50 and IC75 through the cell viability assay, with the aim of confirming that cells incubated with increasing sublethal doses are indeed less sensitive to cisplatin.

After confirming the above, cpR-MCF-7 cells were propagated in parallel with wt-MCF-7 cells to obtain 3 P100 boxes, ensuring they all had the same density, cell passage, and seeding time. This allowed individual RNA extraction, and these RNA samples were sequenced. With the remaining cells, a pool was created, and several tubes were frozen for subsequent functional assays, RNA validation, conditioned medium collection, or vesicle isolation. At each point, three vials (biological replicates) are thawed, and depending on the experiment, it is done in triplicate or more (technical replicates).

Comments 3. In figure 3, explain if the genes that were validated were specifically chosen and the rest could not be validated (and note those as well).

Response 3: As in all transcriptome analyses, if a cutoff greater than 2 is applied, a group of genes is typically selected for validation. These genes can be chosen randomly, based on their fold change, or selected with a focus on specific signal transduction pathways or cellular processes of interest. From this group, at least 80% of the genes must match in validation through another technique, such as RT-PCR, in terms of the modification, to consider the experiment and bioinformatic analysis valid. If the cutoff is lowered to less than a fold change of 2, the number of validated genes must be increased. If validation results in less than 80%, the bioinformatic analysis process is first checked, and if everything is correct, then the experimental design is scrutinized. It is important to consider that there were no experimental groups when the transcriptome analysis was conducted. In such cases, a comparison is made with a component plot that may not clearly separate experimental groups or may exhibit significant variation among experimental samples, such as when analyzing patient samples with intratumoral and intertumoral heterogeneity among patients, as well as differences caused by transportation from the operating room to the RNA extraction site. In the case of cell culture with more controlled models, care must be taken during extraction to eliminate potential biases due to circadian cycles and to ensure the cell density of experimental groups is consistent.

In this specific case, all nine proposed genes for validation were validated at 100%. The selection cutoff for validation was based on genes belonging to the top differentially expressed genes (DEG) or those involved in vesicular or cellular communication processes. When analyzing other signal transduction pathways, additional RNAs are validated, perhaps with a greater emphasis on protein levels to determine if differences in RNA can be translated into protein and functional activity.

Coments 4.MTS and MTT assays do not measure viability directly but metabolic activity. Unless viability was measured independently using other means, this should be corrected in the text.

Response 4: This observation was corrected in the manuscript. While it is an indirect form, the treatment does not specifically damage the mitochondria. This clarification was made in the results section.

Comments 5. In figure 5 C, it is unclear what the identity of the bars are, but assuming that the upper bar corresponds to the cpR cells and the number of vesicles (and not the quality, size, content) is relevant for the observed effect, the authors should try to dilute the particles to equivalent levels as in normal cells and see if the effect disappears. Ideally, the authors should titrate the conditioned medium (in the presence of Cisplatin) of both wt and cpR cells to see if the effects can be simply explained (or correlate) with vesicle numbers. Ideally, the authors should do some minimal molecular characterization (Western blot, Mass spec) of those vesicles to support that conclusion (that the quantity and not the quality is important).

Response 5: The characterization of extracellular vesicles was conducted by western blot using two markers, CD9 and CD63, with antibodies. Briefly, vesicles were isolated using the exoRNeasy Serum/Plasma Maxi Kit (QIAGEN, NV, USA, cat. no. 77064), based on a centrifugation column using membrane affinity binding to extracellular vesicles. Subsequently, proteins were extracted, confirming the proper isolation of extracellular vesicles (EVs) and their presence in conditioned medium through western blot. Thus, we can conclude that the conditioned medium contains extracellular vesicles positive for CD9 and CD63. This likely indicates that, either due to the concentration and/or content of EVs, there is a protective effect against cisplatin-induced cell death in co-cultured cells with conditioned medium from drug-resistant cells.

The afore mentioned results have been included in the supplementary materials section and are also mentioned in the results and discussion. The dilution assays of vesicle concentration seem intriguing as they mimic the scenario where vesicles are present in the bloodstream and various fluids in an organism. Unfortunately, we are still pending completion of this assay due to the slow growth of cisplatin-resistant cells, a persistent limitation in our study. However, we plan to conduct these assays in the future, along with sequencing the transcriptome of the vesicles to analyze which RNAs from resistant cells are transported by vesicles that may influence the transfer of a resistant phenotype to neighboring cells or the conditioning of distant tumor microenvironments during metastasis preparation.

This is of great relevance and current interest, as it could provide evidence of new clinical markers. RNAs contained in vesicles have the potential to be excellent biomarkers, such as liquid biopsies, detectable in peripheral blood (non-invasive test). Being protected within vesicles helps maintain higher concentrations of molecules compared to when they are freely circulating in the blood, increasing the likelihood of avoiding degradation. In this regard, we are already working on the analysis of vesicles from both resistant breast cancer cells and samples from patients in blood and saliva. In conclusion, the proposed experiment to determine the minimum concentration at which vesicles could have an effect is crucial, and we consider it fundamental for the next objectives in this area.

-Coments 6. Since in figure 5A, the authors are using conditioned media, to be able to attribute the effects to vesicles, and not to other soluble factors present in the media, the protective effects from media where vesicles are removed by physical means, should also be shown. Ideally, also, purified vesicles from the media should be added to non-conditioned media and being able to reproduce the effect seen in 5A.

Response 6: Much of this question has been addressed in the previous point. It is a crucial aspect, but due to time constraints, we have decided to incorporate it in the next phase of the project. We hope the reviewer allows us to postpone this assay and include it in the subsequent article.

Minor points

Please see attached text for edits.

We appreciate the criticisms and suggestions from the author, which have helped improve the manuscript and prompted reflection on various aspects of the project overall. We hope the manuscript has now reached the necessary level for publication in the journal, and we are excited to continue the deeper analysis of the resistant phenotype in breast cancer cells and the role of extracellular vesicles.

Round 2

Reviewer 1 Report

Comments and Suggestions for Authors

Authors responded to the comments.

Author Response

The reviewer did not request any new changes.

Reviewer 2 Report

Comments and Suggestions for Authors

Thank you for the revision and the responses.

However, some comments are unaddressed or still unclear in the responses from the authors.

The comment below was implicitly addressed, but the revised manuscript is not updated in accordance with the response.

“Regarding the study material preparation, in order to generate cpR-MCF-7 cells, the authors exposed the wild-type MCF-7 to the drug at various levels of concentration. However, it is unclear at which concentration level the crP-MCF-7 were identified and extracted. If cpR-MCF-7 cell lines were extracted from various levels of drug concentration, multiple cell lines should be extracted from each concentration level to ensure enough number of biological replicates for the differential expression analysis.”

This comment below is not addressed.

“In Figure 5A, the 6th bar from the left (or the rightmost bar) clearly has two clusters of samples, with one cluster having similar viability as the samples in the 4th bar. However, the statistical test between the 4th bar and the 6th bar did not seem to have accounted for this clustering.”

Both comments are crucial in this manuscript as they are directly related to the study materials and the interpretation of the results.

The last two paragraphs of Introduction are repeat of two previous paragraphs. Also, a large chunk of text in Introduction of the previous version seems to have disappeared in this revision.

Are both X-axis and Y-axis of Fig 1D for log fold-change? It doesn’t make sense if that’s the case.

According to Fig 1D, it appears the cut-off of log2(FoldChange) seems to be >5 or <-5. But, the main text says that the cut-off is >1 or <-1. Please clarify.

Also, ‘Fold Change (FCh) ≥ ± 2 (Log FCh ≥ ± 1)’ in the sentence below does not make sense.

“We applied specific cut-off values: Fold Change (FCh) ≥ ± 2 (Log FCh ≥ ± 1)...”

It should be “|Fold Change (FCh)|>2” and “|log2 FCh| >1”

Fig 2B, perhaps ‘mRNA’, instead of ‘RNAm’?

Regarding Fig 3, description of Fig3 A, B, C and D in the figure legend and the main text do not seem correct.

In the second line of page 4, “all the too” -> (perhaps) “all the tools”?

The last sentence of Discussion needs references.

Figure 2B is redundant, as it has the same information as Fig 2C.

In Figure 3, the table parts should be separated out as table(s).

‘Table 1’ is not revised.

Regarding “Response 12”, I am not sure what ‘normalized groups’ and ‘all samples’ mean.

Fig 5A seems to have both the previous and the revised version of the bar plots. I guess this has something to do with the “Response 12” in the author’s responses. However, I can’t comprehend why the number of dots are different and why they have different %viability ranges?

Comments on the Quality of English Language

minor spelling errors.
